# Chemical Overview of Gel Dosimetry Systems: A Comprehensive Review

**DOI:** 10.3390/gels8100663

**Published:** 2022-10-17

**Authors:** Micaela A. Macchione, Sofía Lechón Páez, Miriam C. Strumia, Mauro Valente, Facundo Mattea

**Affiliations:** 1Departamento de Química Orgánica, Facultad de Ciencias Químicas, Universidad Nacional de Córdoba, Av. Haya de la Torre esq. Av. Medina Allende, Córdoba X5000HUA, Argentina; 2Centro de Investigaciones y Transferencia de Villa María (CIT Villa María), CONICET—Universidad Nacional de Villa María, Arturo Jauretche 1555, Córdoba X5900LQC, Argentina; 3Instituto de Investigación y Desarrollo en Ingeniería de Procesos y Química Aplicada (IPQA), CONICET—Universidad Nacional de Córdoba, Av. Vélez Sarsfield 1611, Córdoba X5000HUA, Argentina; 4Instituto Enrique Gaviola (IFEG), CONICET—Universidad Nacional de Córdoba, Av. Medina Allende, Córdoba X5000HUA, Argentina; 5Centro de Excelencia de Física e Ingeniería en Salud (CFIS) y Departamento de Ciencias Físicas, Universidad de la Frontera, Temuco P.O. Box 54-D, Chile

**Keywords:** three-dimensional dosimetry, Fricke, polymer gel dosimetry, nanoparticles, radiotherapy

## Abstract

Advances in radiotherapy technology during the last 25 years have significantly improved both dose conformation to tumors and the preservation of healthy tissues, achieving almost real-time feedback by means of high-precision treatments and theranostics. Owing to this, developing high-performance systems capable of coping with the challenging requirements of modern ionizing radiation is a key issue to overcome the limitations of traditional dosimeters. In this regard, a deep understanding of the physicochemical basis of gel dosimetry, as one of the most promising tools for the evaluation of 3D high-spatial-resolution dose distributions, represents the starting point for developing new and innovative systems. This review aims to contribute thorough descriptions of the chemical processes and interactions that condition gel dosimetry outputs, often phenomenologically addressed, and particularly formulations reported since 2017.

## 1. Introduction

Radiation therapy has been used to treat tumors and cancer diseases for more than one hundred years, only being surpassed by surgery in adult cancer treatment [1]. This treatment modality uses ionizing radiation, mainly high-energy particles, such as X- and γ-rays, electrons, and, to a lesser degree (but with an increasing trend), protons and carbon ions, to selectively kill tumor cells in patients with cancer. Routine radiotherapy treatments need dose-verification protocols and dedicated quality assurance programs to be implemented [2]. To this end, suitable dosimetry systems capable of end-to-end 3D dose verification are required. The typical complete implementation of gel dosimetry is depicted in Figure 1.

Technological advances during the last 25 years have had a huge impact on the delivery of accurate and complex three-dimensional radiation dose distributions in radiotherapy. Technologies such as multi-leaf collimators, together with intensity-modulated beams, have given rise to treatments such as intensity-modulated radiotherapy (IMRT) and volumetric modulated arc therapy (VMAT), which, in combination with optimized treatment planning systems (TPSs), are currently used as the standard planning of the optimal configuration for a specific patient or case [4]. Additionally, image-guided technology, either computed tomography (CT) or magnetic resonance imaging (MRI) [5] combined with IMRT (Image-Guided RadioTherapy (IGRT), MRI-based IGRT, or Magnetic Resonance-guided Radiotherapy (MRgRT)), have been significantly improved and can be used to enhance the precision of treated volumes, reducing healthy tissue irradiation, and have the possibility of correcting the beam position in near-real-time if the tumor moves. Figure 2 shows a scheme of a primary target volume (PTV) near a sensitive organ at risk (OaR) and how new radiotherapy technology improved the dose distribution.

Modern radiotherapy treatments require novel quality assurance methods, must be able to determine highly complex dose distributions with sharp dose gradients, and must have high resolution. Gel dosimetry represents one of the few—if not the only—“true” three-dimensional dosimetry system with an inherent resolution only limited by the analytical method used to read out the response [6]. Moreover, most gel dosimetry formulations are radiological tissue-equivalent, and their response is independent of the radiation orientation, dose rate, and quality [2,7]. In addition, they can be modified and designed for specific purposes or treatments, such as those including the use of nanoparticles, specific drugs, or chemicals. The optimal traditional properties for a dosimetry system, such as reliability, reproducibility, accuracy, and traceability, have not drastically changed during the last few years and have been thoroughly reviewed and discussed elsewhere [2,8]. However, new specific requirements and demands, such as real-time (4D) resolution and improved Linear Energy Transfer (LET) independence, emerged during the development of technologies such as MRgRT, hadron therapy, and Boron Neutron-Capture Therapy (BNCT). For instance, the challenges derived from modern techniques, such as MRgRT, were preliminarily summarized by De Deene [2], while the performance and drawbacks of gel dosimetry for hadron beams, mainly their sensitivity quenching, have been reported by other authors [9,10,11]. Although the traditional characteristics of most used gel dosimetry systems have been properly defined and optimized during the last few decades, it is worth mentioning that the formulation and design of specific gel dosimeters involve organic and inorganic chemistry principles, physics interactions, and analytical techniques that generally exceed the capacity of most of the personnel in typical clinical radiation facilities.

Several review articles have been published in the last few years; for example, Marrale and d’Errico [12] reported thorough and complete analyses of the use of ferrous sulfate-based dosimetry systems, including not only key information on the formulation, synthesis, and read-out of the dosimeters, but also the effect of the gelation temperature and purity of reagents on the reproducibility, response, and dose stability due to ion diffusion or spontaneous oxidation. The authors briefly described the most recent studies on these types of dosimeters, such as reusable Fricke gel dosimetry, for example. Moreover, the use of polyacrylamide gels for ionizing radiation dosimetry was also reviewed by Marrale and d’Errico, with a special focus on clinical applications of the different polymer formulations. In addition, the main scanning techniques were described and compared for Fricke and polymer gels. De Deene [2] recently reported a full review of the historical advances in gel dosimetry, highlighting three main groups: (*i)* Fricke, (*ii)* radiochromic, and (*iii)* polymer gel dosimeters. The state-of-the-art of these three groups was carefully reported, and personal perspectives were presented, focusing on the ongoing challenges for high-performance dosimetry systems. Furthermore, a brief revision of hydrogel-based dosimeters for measuring ionizing radiation doses was recently published by Zhang et al. [13]. The authors classified the dosimeters into the following categories: (*i)* polymer hydrogel, (*ii)* Fricke hydrogel, (*iii)* radio-chromic, (*iv)* radio-fluorescence, and (*v)* NPs-embedded dosimeters, focusing on the main features and drawbacks of their use for routine clinical applications. The sensitivity, accuracy, and dose resolution were specifically considered for each dosimeter.

Although these review articles provide a detailed overview and revision of the historical developments and discussions of the main characteristics and technical elements in each gel dosimetry system, to our knowledge, the interactions between the different components of gel dosimetry have been partially discussed from a chemical point of view. Moreover, the motivations supporting this review rely on the scarce availability of comprehensive descriptions of the chemical processes and interactions that condition gel dosimetry outputs. Often, in the available literature, the effects that arise with modifications and adaptations of suitable dosimetry formulations are empirically addressed by reporting phenomenological outcomes, but sometimes without a robust explanation of the underlying chemical processes. For this reason, this review aims to provide some fundamentals and chemical descriptions relevant to gel dosimetry, and particularly to the materials and formulations proposed during the last five years.

## 2. Brief History of Gel Dosimetry

Complete and extensive revisions of the different proposed and implemented dosimetry systems have been already presented by several authors during the last decade [6,12,14,15] and do not represent the main goal of this review. Nevertheless, a summary must be included to understand the motivation for the novel and innovative systems proposed during the last few years. Two different types of gel dosimetry are typically considered—those based on the oxidation of ferrous sulfate into ferric ions in an acid solution within a gelatin matrix or, more recently, in a polyvinyl alcohol-based matrix, known as Fricke and PVA-GTA Fricke gels, respectively, and those based on polymerization reactions induced by the radicals formed during the radiolysis of water, commonly known as polymer gel dosimetry (PGD).

Fricke gel dosimetry was derived from the initial studies by Fricke and Morse in 1927 [16] and emerged with the use of nuclear magnetic resonance (NMR) to measure the radiation-induced chemical changes while stabilizing the dose information with the aid of a gel matrix [17]. In these dosimeters, ferrous sulfate is incorporated into a gel matrix and the physicochemical change upon irradiation is the oxidation of Fe^2+^ ions to Fe^3+^ aided by the radicals generated in the radiolysis of water by the ionizing radiation. Therefore, the final concentration of Fe^3+^ depends on the absorbed dose and the chemical yield of ferric ions *G*(Fe^3+^). The Fe^3+^ concentration can be measured by spectrophotometry in the ultraviolet region, mainly at 304 nm and 224 nm [17]. The main limitation of these dosimetry systems is the loss of spatial information due to the diffusion of ferric ions within the gel matrix [18]. Many improvements and enhancements have been proposed over the decades, with a special focus on reducing autoxidation processes and the diffusion of ferric ions, and on improving or simplifying the response read-out of the dosimeters. For example, adding an indicator, such as xylenol orange, to an acid solution of ferric ions enhances the optical absorbance in the visible range and reduces the diffusion of ferric ions by forming a metal coordination complex [19,20]. Another example is the use of sodium chloride to increase the reproducibility of MRI readouts in Fricke gels containing organic impurities [21]. A completely different approach was proposed by Chu et al. [22] aiming to reduce the diffusion of ferric ions in Fricke dosimetry by using a different gel matrix. In their study, the authors used polyvinyl alcohol (PVA) cryogels or PVA hydrogels prepared with PVA concentrations of up to 20 wt%, which exhibited ten times lower auto-oxidation rates than gelatin-based Fricke gels, and diffusion coefficients of less than half of the lowest reported value. However, the obtained cryogels had a rubbery consistency, opaque appearance, and expanded upon freezing, limiting their application and readouts for MRI. Then, Marini et al. [23] suggested preparing the gel matrix by the chemical crosslinking of PVA with glutaraldehyde (PVA-GTA). The obtained dosimetry system overcame the limitations of PVA cryogels, exhibiting high sensitivity, a low minimum detection dose, and low diffusion rates. Moreover, great advances have been made by using PVA-GTA formulations [24,25,26,27,28].

The main chemical reactions in these types of dosimetry systems can be summarized in the reaction of ferrous ions and radicals formed upon irradiation or their subproducts in an acid solution.
$$Fe^{2+}+OH^{•\;}\to Fe^{3+}+OH^{-}$$
(1)$$Fe^{2+}+HO_{2}^{•\;}\to Fe^{3+}+HO_{2}^{-}\;$$
$$HO_{2}^{-}+H_{3}O^{+}\to H_{2}O_{2}+H_{2}O\;$$
$$Fe^{2+}+H_{2}O_{2}\to Fe^{3+}+OH^{-}+OH^{•\;}$$

The use of polymers in radiation dosimetry was originally focused on the degradation of the macromolecules upon radiation and the subsequent viscosity decrease [29,30]. In 1958, Hoecker and Watkins proposed a polymerization reaction induced by radiation and a dosimetry system relying on adjusting the sensitivity of the polymerization with dissolved iodine or oxygen [31]. Several decades later, the use of NMR to assess changes during the polymerization of N,N′-methylene-bis-acrylamide (MBA) [32] and acrylamide (AAm) with MBA [33] within a gel matrix set the basis of modern PGDs. In the early stages of polymer gel dosimetry, several issues related to their accuracy and reproducibility for clinical applications appeared because of the inhibitory effects of oxygen in free radical polymerizations, which forced their manufacture, storage, and irradiation under hypoxic conditions [8]. It was only in 2001 that Fong et al. [34] included, in a PGD, a complex formed with ascorbic acid, copper sulfate, and oxygen that could produce free radicals and be used in polymer gel dosimetry under a normal atmosphere, but it required the use of a radical inhibitor, such as hydroquinone, to avoid any polymerization prior to their use. After that, complexation with oxygen was replaced by just ascorbic acid as an antioxidant, and then with a better antioxidant, such as tetrakis (hydroxymethly) phosphonium chloride (THPC), that may also act as a promoter of the polymerization reaction [35]. These new types of material called normoxic polymer gel dosimeters have been widely used for clinical applications, as reviewed by Farhood et al., up to late 2017 [15]. A similar systematic approach to that used by Farhood et al. was followed in the present study to include some recent studies intimately related to clinical applications. The adopted methodology consisted of a meta-search of the titles and abstracts of articles in three different electronic databases, namely Scopus, PubMed, and Web of Science, from the last five years up to 9 October 2022. The search terms included three groups related to “polymer gel dosimetry”, “type of radiation therapy”, and “type of polymer gel dosimetry”. Further steps consisted of the exclusion of duplicates, manual screening based on the title and abstract, and final selection by full-text analysis. The extracted data from the selected articles are summarized in Table 1.

New trends, compositions, and variations in most gel dosimetry systems used have been presented over the last twenty years, aiming to provide dosimetry systems for specific readout techniques, and mainly those implemented in standard-equipped linacs, such as the case of the massive incorporation of conventional CT, kilo- and mega-voltage cone-beam CT into treatment rooms for near real-time monitoring. This approach was pioneered in 1999 [53], extended to real-time 2D IGRT to detect gross motion and correct the treatment, and then evolved into 3D real-time IGRT [54]. In this framework, gel dosimetry specially designed to exhibit higher sensitivity for X-ray CT and the optimization of CT imaging protocols were proposed in 2000 [55] and improved over the years. For example, Jirasek et al. [56] used a highly concentrated NIPAM-based dosimetry system containing 30% isopropanol as a cosolvent to increase the solubility of the reactive system from the typical 6% used in PGDs up to 20 wt% with 50% crosslinkers. Chain et al. [57] proved that the solubility of the commonly used crosslinker MBA can be increased from 3% in water to 5.5% in NIPAM solutions, thereby achieving high total concentrations (~19.5 wt%) suitable for X-ray CT readout with only aqueous solutions. Recently, Javaheri et al. [58] reported an optimized CT protocol that was able to measure CT number changes in NIPAM dosimeters with typical mass concentrations (3% NIPAM and 3% MBA) for irradiations with doses from 2 to 8 Gy. Jirasek et al. [59] studied gels with 14.5% NIPAM and 4.5% MBA for near-real-time readouts in Linac-integrated kV cone-beam CT. The obtained results proved that, with a post-irradiation time of 20 min, 90–93% of the radiation-induced CT number change takes place if a proper number of acquisitions, image averaging, and filtering are used. The authors also demonstrated that the accuracy and precision can be improved to 2–4%, and 3D gamma tests against treatment plans yielded values above 90% with the proposed setup.

## 3. Readout Techniques

The readout techniques used in gel dosimetry have a great influence on their final application; in most cases, the available analytical methods in clinical facilities are favored. Thus, efforts have been made during the last few decades to apply MRI [60,61] and even use onboard MRI scanners included in modern MRI-based IGRT [62], X-ray computed tomography (CT) [58,63,64], or optical computed tomography methods (OCT) [65,66,67] to deliver 3D response distributions in gel dosimetry systems in clinical applications. Raman spectroscopy represents another technique widely used in gel dosimetry, but mainly for research purposes, as it provides quantitative information on the consumption of each reactive species without modifying the dosimeters and is capable of obtaining 2D distributions and even 3D distribution at the microscale [68,69,70].

## 4. Radio Physical Processes

Energy deposition in matter (*E_dep_*) arises from the different interaction mechanisms of ionizing radiation, which depend on the radiation type [71]. Three main mechanisms apply to electromagnetic radiation, namely the photoelectric effect, Compton effect, and pair production, whereas the main interaction mechanisms for charged particles are Bremsstrahlung, annihilation, and inelastic and elastic collisions [72]. The main interaction mechanisms for neutrons are nuclear reactions, neutron capture, and elastic and inelastic scatterings [73]. Given a specific material, the relative importance of each interaction mechanism is characterized by the corresponding double differential cross-section (d*σ*/dEd*Ω*) that depends on the kinetic energy (*E*) and scattering solid angle (*Ω*) of the particle passing through, while the total cross section (*σ*) can be directly obtained by integrating d*σ*/dEd*Ω* over E and *Ω*. Owing to ionizing radiation passing through matter, dense ionization sites (spurs) are formed. Therefore, noticeable inhomogeneous spatial energy depositions arise. Indirect ionizing radiation, such as photons and neutrons, achieve energy deposition by means of the secondary charged particles created within the irradiated material; thus, understanding the interactions between charged particles, mainly electrons, stands as the key issue. The linear energy transfer represents the mean rate of energy loss by a particle per unit of path length traveled in matter, and it changes as the charged particle kinetic energy varies while progressing on the track as depicted by the Bethe formalism [74,75].

Then, ions, radicals, and excited electrons (e^−^_th_) are locally and inhomogeneously produced, originating dense ionization sites in a very short time scale. Considering the orders of magnitude of atomic/molecular dimensions (∼10^−10^ m) in a relativistic framework, charged particles require ∼10^−16^ s to travel through. Moreover, according to the Heisenberg uncertainty principle, energy loss of ∼1 to 10 eV, as representative threshold values for the required ionization energy, corresponds to discernible times of t ≳ 10^−16^ to 10^−15^ s, respectively. These short-term radiation–matter interaction events initiate ion–radical production, triggering subsequent chemical reactions, as well as their corresponding further biological effects. In summary, three stages, broadly defined and overlapped, can be recognized and characterized by different time scales: (*i*) physical, (*ii*) chemical, and (*iii*) biological. Once a primordial radiation–matter interaction event occurs, radiophysical and radiochemical processes begin. Four chronological phases may be identified in aqueous systems [76], such as: (*a*) thermalization: ionization electron slows down due to interactions with molecules. The initial kinetic energy *E* of the ionization electron is then transferred to matter by knocking the orbital electrons of those molecules, producing excitations and ionizations. This process continues until the electron kinetic energy *E* drops to a few eV. (*b*) Localization: thermalized electrons with remaining kinetic energy of a few eV occupy the water conduction band that is mostly empty due to its dielectric properties. Thus, thermalized electrons move through the water for about 10^−13^ s to find a localized state in the remaining energy of a few eV, i.e., an Eigenstate with reduced spread out wavefunction, commonly a “shallow trap”. (*c*) Solvation: The electrical interactions between the trapped (ionization) electron and surrounding molecules reduce the remaining electron kinetic energy, obtaining a solvated electron in about 10^−13^ s. (*d*) Recombination: following solvation, the (ionization) electron combines with a positively charged ion. The elapsed time depends strongly on the surrounding positive charge distribution. To this end, lone protons play a relevant role in aqueous systems.

At a cellular level, the most important effect of ionizing radiation is DNA damage, limiting the ability of cells to replicate. Healthy cells can repair most of this damage, while cancer cells do not, achieving a targeted and effective treatment [77].

From a chemical point of view, ionizing radiation involves several processes and species that occur between the atomic physical interactions and the biological repercussions, known as radiochemistry, which is mainly due to water radiolysis in aqueous systems. This stage involves a variety of reactive intermediates close to the ionization site, which give rise to small clusters of excited and ionized molecules with •OH, H•, H+, and e^−^_th_ as the main species. Then, the solvation and diffusion of these species and radicals and their recombination takes place, leading to a homogeneous regime where the radicals e^−^_aq_, •OH, H•, and HO_2_•, and molecular products H_2_ and H_2_O_2_ are predominant, and can lead to reduction (e^−^_aq_, H•) or oxidation reactions (•OH, HO_2_•, H_2_O_2_), with the hydroxyl radical (•OH) being mainly responsible for the oxidation of biological molecules. A detailed description of the time scales, chemical species, and physicochemical processes involved in the radiolysis of water can be found in [78], and it is essential to understand the nature and principles of gel dosimetry systems. Figure 3 summarizes all of the above-mentioned mechanisms.

## 5. Radiochemical Modeling of Polymer Gel Dosimetry

Chemically, the concept of polymer gel dosimetry is based on the initiation of a polymerization reaction in a mixture of vinyl monomers and crosslinking molecules dissolved in a gel matrix by the radicals formed during water radiolysis. Like every polymerization reaction, there are three main steps: (*a*) initiation, where radical monomers are formed; (*b*) propagation, where the polymer chain grows and forms a crosslinked network; and (*c*) termination, which encompasses several reactions where radicals are consumed or combined with other radicals. A full description of the main reactions involved in polymer gel dosimetry can be found elsewhere [8,79], and together with the kinetics and phase separation in the dosimeters, are essential to model their dose–response. For example, Fuxman et al. [80] modeled the copolymerization of AAm and MBA in an aqueous gelatin matrix considering a spatially uniform radiation dose distribution as the primary approach toward developing a valuable model for 3D radiation dosimetry. The proposed model considered ring cyclization as the main mode for intramolecular crosslinking and made several assumptions to simplify the complexity of the system, such as considering only two types of radicals, ones with high mobility (small molecules) and other ones that are completely immobile (long crosslinked molecules), neglecting the consumption of radicals by impurities or dissolved oxygen, considering the same reactivity of the vinyl bonds in AAm and MBA, disregarding any temperature changes caused by polymer precipitation, insignificant volume changes due to changes in density during polymerization, no direct action of radiation on the monomers or gelatin, only crosslinked polymers precipitating into a polymer phase, the reactions in the aqueous phase being controlled by the kinetics of the reaction but the ones in the polymer phase being controlled by diffusion, etc. These simplifications evince the complex phenomena involved in polymer gel dosimetry, even for the case of an anoxic system, such as PAG, with no additives or improvements. Nonetheless, the model was able to predict vinyl group conversion during irradiation and in the post-irradiation stage, the concentration of unreacted pendant vinyl bonds, the number of cyclized groups and crosslinks in the precipitated polymer, and the temperature changes associated with the polymerization, yielding results comparable to experimentally measured values. Two years later, the same authors improved the model to account for a spatially nonuniform dose distribution [81] by considering a differential volume wherein the continuous aqueous phase coexisted with small, precipitated polymer particles with the same distribution of chemical species. This model considers the diffusion of monomers and radicals within the aqueous phase because of the concentration gradients created by the non-spatially uniform irradiation and represents a better description of the complex phenomena in a real polyacrylamide irradiation-based dosimetry system. One step closer to a real scenario was studied by Koeva et al. [82] by accounting for oxygen contamination or the presence of a radical inhibitor, such as monomethyl ether hydroquinone, in PAG and NIPAM dosimeters. The authors proved that the presence of the typical inhibitor concentration in commercial acrylamide-based monomers had a significant effect on the dose–response only if oxygen was present in the gel. These models were used to study specific dosimetry applications, such as the depth–dose–response in PAG irradiated with ^60^Co γ, 6 and 15 MV X-ray photon beams, and 6 to 20 MeV electron beams [83], observing small inaccuracies in the depth–dose calibration curves from PAG due to monomer diffusion, long-lived radicals, dose rate effects, and temperature effects on the reaction rates of the polymerization reactions. Nasr et al. [84] also used the described model to simulate HDR and LDR brachytherapy in PAG dosimeters by using a spherical seed of ^192^Ir implanted temporally and an ^125^I seed implanted permanently, respectively. The authors of these models recognized the necessity of including the reactions and processes related to the antioxidants used in normoxic polymer dosimetry, especially for THPC as it is widely used in many dosimetry formulations. However, they concluded that THPC is involved in many chemical reactions within polymer gel dosimetry that are not fully described yet, and further research is required to properly account for its effects.

## 6. Operational Properties of Gel Dosimetry

Gel dosimetry covers a wide variety of materials that exhibit quantitative changes in their chemical or physicochemical properties with the interaction with ionizing radiation and that have the capacity for accurately, precisely, and reproducibly recording these changes as a function of the absorbed dose with 3D spatial resolution [2,8]. Their main application is providing quality assurance tools for radiotherapy to ensure the proper delivery of precise and well-known dose distributions conforming to the PTV while minimizing the dose in the surrounding healthy tissue and organs. Modern radiotherapy treatments, such as IGRT, MRgRT, or theranostics applications based on metallic nanoparticles, demand advanced dosimetry performance for challenging requirements commonly found in mixed fields, where different dosimetry contributions as well as internal/external effects, must be assessed. Gel dosimetry offers the outstanding flexibility of being chemically and isotopically adjustable to include these effects, thereby constituting a promising tool to provide a fair description of the dose distribution in complex radiotherapy. Some of the most relevant and practical characteristics of gel dosimetry related to their ability to register 3D dose distributions are summarized in the following sections, along with the corresponding analysis and interpretation in terms of chemical fundamentals. Figure 4 presents a qualitative description of the most relevant operational properties of gel dosimetry.

### 6.1. Sensitivity, Dynamic Range, and Minimum Detectable Dose

For most gel dosimetry systems, there are two main contributions defining the response of the dosimetry system—on one hand, the physical and chemical processes that take place after the radiation-induced radiolysis of water, and on the other hand, the analytical method used to correlate physicochemical changes within the material with the dose values. For example, if the minimum detectable dose, also known as dose threshold, is considered in a specific gel dosimetry system, such as Fricke gel, the chemical reduction of Fe^2+^ into Fe^3+^ begins with very low dose values, being limited only by the analytical readout technique used to measure the concentration of Fe^3+^ in the gel. Babic et al. [85] studied the effects of varying the xylenol orange supplier and concentration, ferrous sulfate and gelatin concentrations, and the wavelength used in the spectrophotometric readout of the irradiated materials. In their study, the minimum threshold changed with the xylenol orange purity (or supplier) and concentration, but not by removing the gelatin matrix. For that reason, the authors attributed the threshold differences to the formation of different complexes between Fe^3+^ and xylenol orange, depending on their relative concentration (Fe^3+^_i_:XO_j_ complexes), combined with the differences in the spectral curve for each species. Then, if Fe^3+^_2_:XO_1_ is favored because of significant autoxidation processes, a pre-exposition to radiation, or the incorporation of Fe^3+^ to the formulation, longer wavelengths should be used (i.e., Fe^3+^_2_:XO_1_ has a maximum absorbance at 580–590 nm, yellow–orange range) instead of the typical wavelength used for standard Xylenol Fricke gel dosimetry (543.5 nm, green). Furthermore, the sensitivity of gel dosimetry, characterized by the slope of the response–dose curve, changed significantly by using different read-out wavelengths.

Moreover, several readout techniques can be used to measure the induced changes in the dosimeters and, consequently, the dose–response or sensitivity. For example, Huang et al. [86] compared optical laser scanning, solid-state ^1^H-NMR and ^13^C-NMR, and FT-Raman spectroscopy to measure the response of NIPAM-based dosimeters irradiated with doses up to 30 Gy. In their study, a linear response was observed up to 15 Gy with optical methods, while exponential trends were reported for the consumption of monomers measured by FT-Raman spectroscopy, and they only presented results obtained by NMR, where the intensities of the carbon double-bond (in the unreacted monomers and crosslinking agents) were reduced to half.

The dynamic range of the proposed dosimetry formulation, measured as the useful dose range extracted from the readout of the irradiated dosimeters, should depend mostly on the concentration of the sensitive species and on the concentration or crosslinking degree of the gel matrix, and secondly on the propagation and termination kinetics of the polymerization reaction. Extreme variations in the dose rate inducing significantly different oxygen and ion-radical environments may also affect the gel dosimetry response, to some extent. Additionally, the readout techniques are limited by their own analytical sensitivity or saturation effects, such as optical density measurements. Furthermore, in some cases, the optical properties of the synthesized hydrogels depend on their crosslinking degree achieved during irradiation, which, in turn, depends on the relative concentration of monomers and crosslinking agents in the dosimeter. For example, Mattea et al. [87] studied the radiation-induced polymerization of itaconic acid crosslinked with MBA in a gelatin matrix by FT-Raman spectroscopy and optical methods, proving that, if oxygen was present in the dosimeter, the relative incorporation of itaconic acid in the hydrogel changed, thus changing the structure of the obtained hydrogel together with its optical density, thereby leading to different sensitivity values. From a chemical point of view, it is not evident that the optical properties or mass density of a dosimetry material should evolve linearly with the induced polymerization. The relative reactivity between monomers and crosslinking agents, along with their diffusivity within the gelatin matrix, could significantly affect the molecular conformation of the hydrogel, which may not remain the same between the initial steps of polymerization and the last steps or the post-irradiation polymerization. Some of these properties for gel dosimeters typically used in clinical applications can be found in Table 2 of reference [15].

### 6.2. Stability and Reproducibility

Chemical instabilities in gel dosimeters depend on each specific formulation, as reported by De Deene et al. [88] and Vergote et al. [89]. For polymer gel dosimeters, two main issues have been reported that influence the system stability, which are closely related to the polymerization that continues after the irradiation and to the gelation process [88]. A full technical description of polymer gel dosimetry stability was provided by Ibbott [90], Bayreder et al. [91], and De Deene et al. [92]. Additionally, practical strategies have been proposed to handle the temporal chemical instability by following strict protocols during the sample manufacture, storage, calibration, irradiation, and readout processes [93,94]. As noted by Gambarini et al. [95], the reproducibility of Fricke gel dosimetry can be significantly improved by implementing dedicated tuned analytical methods. Schreiner emphasized other practical issues to improve the reproducibility of Fricke gel dosimetry [18] that are associated with the proper preparation and general managing of samples, remarking that such procedures and the corresponding facilities for preparing reproducible Fricke gel dosimeters are, in turn, considerably simpler than those required for most polymer-based dosimeters. Ibbott [90] and Maryasnki et al. [33] reported that BANG polymer gel dosimeters attained a highly reproducible response up to 8 Gy. Cosgrove et al. [96] reported that PAG systems, as used to verify both simple and complex treatment plans, achieved relative dose distributions that were reproducible and in good agreement with the planned dose distributions.

### 6.3. Energy and Dose Rate Dependency

The output of most gel dosimeters does not depend significantly on the radiation quality when considering high-energy photon and electron radiotherapy beams [7,88] due to the similar *LET* in the MeV energy range for these particles [97]. However, dedicated preliminary characterization is required to use gel dosimetry for applications with non-negligible *LET* differences compared with typical radiotherapy radiation qualities, as occurs for brachytherapy [98,99,100] and radiology [14,101,102]. In order to avoid systematic errors in the dose distribution assessment for low-energy applications, the corresponding gel dosimeter dose–response output needs to be obtained for the specific beam quality, and the relative performance to the reference quality (^60^Co) may serve to correlate with their application in the MeV range. The energy dependence of Fricke gel dosimetry for in vivo measurements in high-dose-rate brachytherapy has been reported by Carrara et al. [101], who investigated photon beam energies below 100 keV. Additionally, Šolc and Sochor [103] reported two Fricke gel dosimeter formulations feasible for low-energy (brachytherapy) applications to assess full relative 3D dose distributions, observing almost full energy independence for photon energies higher than 40 keV and a non-negligible response reduction for photon energies lower than 15 keV. A formulation of the polymer gel dosimeter BANG analyzed by high-resolution laser CT was investigated by Massillon-JL et al. [104] for intravascular brachytherapy, who reported a quite uniform and energy-independent response for low-energy photons in the range of 20–1250 keV. Similarly, an extensive study involving different polymer gel dosimeter formulations, such as BANG, MAGAT, MAGIC, and VIPAR, was reported by Pantelis et al. [105], showing uniform properties within the 0.1 to 10 MeV energy range. In this sense, correction factors of less than 5% were reported for ^125^I and ^103^Pd brachytherapy applications [106], and similar results were reported for MAGAT in radiology (80 to 120 kVp) by Antoniou et al. [107].

It is useful to distinguish between low- (up to ∼2 Gy/h), medium- (up to ∼12 Gy/h), and high (more than ∼12 Gy/h)-dose-rate regimens in clinical applications. Accordingly, selecting an appropriate gel dosimetry formulation for these rate regimes is a key issue. For Fricke gel dosimetry, Chu reported in 2001 [108] that the dosimeters are flexible enough to provide an acceptable dose–response for dose rates varying from a fraction of one Gy/min to tens of Gy/s. Similarly, Šolc et al. [103] reported, in 2012, results from ^125^I and ^192^Ir brachytherapy showing almost full dose rate independence of Fricke gel dosimetry up to 85 Gy/h (∼1.5 Gy/min). VIPAR polymer gel dosimetry outputs were compared by Kipouros et al. [109] irradiating with 6 MV photon and ^192^Ir sources, and they obtained agreements between calibration curves, which were discussed as an indicator for both beam quality and dose independence. The dose rate dependence for PAGAT polymer gel dosimetry was reported by Zehtabian et al. in 2012 [110], demonstrating an almost uniform response, except for very low (∼2 Gy/h) regimes, while a similar trend, i.e., uniform response for dose rates higher than ∼1 Gy/h, was reported for PRESAGE polymer gel dosimeter by Pappas et al. [111]. The BANG polymer gel dosimeter was widely characterized by Massillon-JL et al. in terms of its dose rate dependence using ^60^Co [89] and ^90^Sr/^90^Y beta particles brachytherapy [112], obtaining noticeable dose rate independences that motivated the authors to state that gel dosimetry fully revealed the radial dose function, and they concluded that dose measurements at distances very close to the brachytherapy seed are reliable.

### 6.4. Accuracy and Precision

In 2006, De Deene reported a detailed study addressing issues related to accuracy and precision in gel dosimetry [88], highlighting that these properties can be evaluated both in terms of dose and space, as the latter two are strongly interwoven. It is worth noting that both accuracy and precision in gel dosimetry depend not only on the formulation itself, but also on the analytical technique. Owing to the inherent 3D resolution of gel dosimetry, reporting integral errors, i.e., dose and spatial uncertainties together, has become a common practice. The gamma index, originally proposed by Low et al. in 1998 [98], quantifies both errors in one common parameter, the *γ*-index. Moreover, appropriate accuracy assessment requires “true data” or a gold standard to quantify corresponding differences and deviations. Most practical approaches consist of employing dose distributions obtained by a reliable alternative technique, treatment planning, Monte Carlo, or other experimental devices, as a reference pattern for comparison. Nonetheless, issues arising from different spatial resolutions, sensitive volume, or even differences in basic dose–response outputs need to be overcome to accomplish such comparisons. Detailed descriptions of typical sources of inaccuracy in gel dosimetry, including chemical instability, sample positioning, thermal history, and the calibration process, among others, are available in [88,113].

### 6.5. Water Equivalence

Gel dosimetry implementation in clinics is partially motivated by the quite unique characteristics of this type of dosimeter, where the dosimeter constitutes both the radiosensitive volume (dosimeter) and the phantom. However, the assessment of experimental dose distributions requires a proper representation of biological tissues [8]. To this end, soft tissue is commonly considered, from a practical point of view, radiologically equivalent to liquid water. In general terms, two different materials can be considered to be radiologically equivalent for a certain energy range if they present equal fundamental physical quantities [102], such as the cross-section and stopping power, for instance. For high-energy photon irradiation, most gel dosimetry formulations can be considered as water (soft tissue) equivalent. However, for low-energy applications, such as radiology and brachytherapy, deviations may occur, as reported by Baldock [8], who showed differences within 5% for ^125^I and ^103^Pd, while Ibott stated some concerns regarding low-energy regimens [94] and suggested that MAGAT gel dosimetry appeared, at the time, as the most water-equivalent over a wide range of energies, as previously reported by Hill et al. [114]. Pantelis et al. [105] reported calculations of the effective atomic number (*Z_ef_*) and electron densities of different gel dosimetry formulations, including BANG, MAGIC, MAGAS, and Fricke, obtaining, in all cases, *Z_ef_* ∼ 7.4 and discrepancies in terms of electron density (ρ_e_) within 1% as compared with soft tissue. Analogue results for a low-energy range were reported by Valente et al. in 2018 for NIPAM, PAGAT, itaconic acid (ITABIS), and Fricke formulations in the low-energy range (up to 130 kVp), reporting overall variations with respect to water of less than 3%, as obtained by Monte Carlo simulations [102].

## 7. Chemical and Physical Interactions Present in Gel Dosimetry

### 7.1. Gel Matrix

The ability of dosimeters to preserve the spatial information of a dose can be disturbed by low gel melting points. One of the most used gelling agents in gel dosimetry is gelatin, which is basically the denatured form of collagen. The gelling process of gelatin involves the reorganization of the polypeptide chains after the relaxation of the molecules in hot water when the temperature decreases. In such reorganization, a balance between intermolecular hydrogen bonds and intramolecular bonds defines the matrix architecture and properties, such as mesh and pore sizes, that have a direct influence on the diffusion of species within the gel. The abovementioned balance is susceptible to pH, temperature, and concentration variations [115], and to the presence of additives that could interfere with the hydrogen bonding in the matrix [116]. Therefore, some strategies are used to increase the melting point of gels, allowing their use in warmer environments. For example, Fernandes et al. [117] achieved an increase in the melting point of MAGIC-type gel dosimeters by using formaldehyde, which increased the cross-linking reactions in gelatin molecules while improving the sensitivity and reducing the uncertainty in the NMR characterization. Romero et al. [116] studied the crosslinking of the gelatin matrix with glutaraldehyde to preserve the mechanical properties of the gelatin at higher temperatures without significantly affecting the dose–response in ITABIS dosimeters. Moreover, the same modification was studied by Chacón et al. [118] to minimize the effect of inorganic salts incorporated to enhance the sensitivity of PAGAT dosimeters on the stability and depression of the melting point of gelatin. One of the most relevant matrix modifications may be the crosslinking of polyvinyl alcohol with glutaraldehyde for Fricke-based dosimetry [119]; in this case, the main motivation was simplifying the gelling process of PVA by using a similar approach consistent with that used in glutaraldehyde and gelatin. Moreover, the influence of gelling temperature on this type of matrix was studied by Gallo et al. [26,120]. An extensive and complete revision of these types of dosimeters can be found elsewhere [12].

A special Interaction with the gel matrix, and particularly with gelatin, which is worth mentioning, is its reaction with some monomers, such as methacrylic acid, by a graft polymerization mechanism [8], as occurs in other matrices or polymers, such as rubber [121]. Some other gel matrices have been proposed and used in gel dosimetry. For example, a mixture of agar and gelatin was used with methacrylic acid (MAGT-A) enclosed in a vinyl film to produce deformable and flexible dosimetry systems, which were in agreement with ionization chamber measurements within 3%, except for the surface region [122]. Maeyama et al. [123] used a commercial electrostatically crosslinked organic–inorganic gel matrix called AQUAJOINT^®^ that had no C–C bonds, could be prepared without heating, exhibited high elasticity, and resisted compression and heating up to 120 °C. The authors modified the dosimeter VIPET based on NVP by replacing the gelatin with the new gel matrix and obtained a 2.5-fold increase in sensitivity compared with the standard VIPET and stability for at least 56 days at low temperatures. The same authors proposed using clay nanoparticles called Laponite XLG to modify the gelatin matrix of Fricke gel dosimeters, achieving complete suppression of the diffusion of ferric ions after irradiation by inducing the adsorption of Fe^3+^ cations into the interlayer of nano clay by an ion-exchange reaction. Surprisingly, the authors also observed an independent response with the LET, possibly due to the compensation of the typical decrease in the sensitivity of Fricke-based dosimetry with the LET, with the chain reactions promoted by the molecular oxygen in the new formulations. Jaszczak et al. [124] studied the substitution of gelatin with a matrix composed of poly(ethylene oxide)-block-poly(propylene oxide)-block-poly(ethylene oxide) (Pluronic F-127) in MAGAT, PAGAT, NIPAM, and two dosimeters based on NVP (VIP and VIC). The authors reported that Pluronic F-127 drastically changed the solubility of methyl methacrylate, resulting in a viscous, soft, gel-like inhomogeneous product, possibly due to the rapid micellization and gelation of Pluronic. They also observed that the solubilities of the main components of VIC were significantly decreased when Pluronic was used and limited its manufacturing. NIPAM-based dosimeters exhibited a negligible dose–response, while PAGAT and VIP showed better sensitivities and a better dynamic dose range. It becomes clear from the overall study that using a new gel matrix could generate new interactions with the reactive species that may cause melting point depression of the gel, inhibit the polymerization reaction, alter the solubility of the main components in dosimetry, or improve the response and performance in some cases. Some of the main characteristics and examples of different gel matrices are presented in Figure 5. 

### 7.2. Cosolvents

The use of cosolvents typically involves adding organic solvents with good miscibility with water to the gel dosimetry system in order to increase the solubility of reactive species (in particular for the crosslinking agent) [125] and consequently increase their local concentration, achieving higher conversions and responses in the dosimeters. One clear example of this approach is the use of cosolvents to improve the solubility of the crosslinking agent MBA to values higher than that of water (around 3 wt%). Koeva et al. [126] studied the use of several cosolvents with polar and nonpolar functional groups, such as glycerol, isopropanol, *n*-propanol, sec-butanol, *n*-butanol, methylethyl ketone, and ethyl acetate, to increase the concentration of MBA in a NIPAM dosimeter with a 5 wt% gelatin concentration. In an initial step, the authors observed that *n*-butanol, methylethyl ketone, and ethyl acetate at concentrations higher than 10 vol% were not suitable for preparing the gelatin matrix due to their partial miscibility with water. From the remaining solvents, *n*-propanol and sec-butanol yielded cloudy gels and were not considered in the study, and, in some cases, phase separation occurred over time after the manufacturing process. NIPAM dosimeters containing 5 to 7 wt% MBA and the same amount of monomer were prepared with 10 vol% isopropanol and glycerol. A remarkable increase in sensitivity was achieved with these formulations, especially with isopropanol, where a 39% increase was observed compared with the typical NIPAM formulation. In addition to permitting a higher concentration of reactive species, isopropanol is a chain transfer agent for free-radical polymerization that may influence the molecular size and conformation, and consequently the size of the polymer particles that precipitate within the gel network. With a similar approach, Kozicki et al. [127] used 5 to 30% isopropanol or 4 to 20% tert-butanol as cosolvents to increase the concentration of MBA in VIPAR gel dosimeters prepared with 17 wt% N-vinylpyrrolidone (NVP). The results showed that promising and stable polymer gels were obtained by using tert-butanol, but polymerization inhibitors had to be incorporated to reduce the auto polymerization of MBA and copolymerization of NVP with time. The final formulation, which was named VIC, exhibited a sensitivity 2.2 times higher than that of standard VIPAR^nd^, promising stability and suitability for CT readout.

Acetone was also proposed and studied as a cosolvent to enhance the performance of PGDs. For example, Rabaeh et al. [128] studied the solubility of N-(isobutoxymethyl) acrylamide (NIBMA) monomers in NIBMAGAT dosimeters. In this case, the authors were able to double the total concentration of comonomers (NIBMA and BIS) from 4 wt% to 8 wt% by incorporating 20 vol% of acetone and 20% of glycerol as cosolvents, which also resulted in a sixfold increase in the sensitivity value within a linear dose range of 0 to 10 Gy measured by NMR.

Cosolvents can also be used as a strategy to reduce the diffusion of species, aiming to preserve a stable dose distribution during and after the irradiation. For that purpose, Rabaeh et al. [27] studied the effects of the incorporation of dimethyl sulfoxide (DMSO) as a cosolvent, which also acts as a free radical scavenger, on the performance of transparent Fricke methylthymol blue dye PVA-GTA. The authors observed a slight decrease in the sensitivity, but better temporal stability and optical properties in the dosimetry system. The effects of DMSO on gel dosimeters are, on one hand, related to the stabilization of Fe^3+^ and minimization of autoxidation processes, and on the other hand, involved in the interaction with the PVA-GTA matrix, thereby changing the optical properties of the gel and its tensile strength. The chemical reactions involving DMSO are first centered on the reaction with OH^•^ radicals, competing with ferrous ions in the redox reactions and thereby minimizing autoxidation, but reducing the overall sensitivity to ionizing radiation. Secondly, the DMSO radicals could also reduce the ferrous ions into ferric ions and, finally, a less frequent byproduct of the reaction between hydroxyl radicals with DMSO is the formation of formaldehyde by a Russell reaction mechanism. Formaldehyde could act as a crosslinking agent in PVA networks [129], explaining the increase in the tensile strength and the decrease in the diffusivity of ferric ions; however, temperatures of 50 °C are required. Thus this effect would be less probable during irradiation. A more plausible explanation for the changes in the diffusion of ferric ions could be related to the sharp increase in the hydrophobicity in a water–dimethyl sulfoxide (DMSO) binary mixture at low DMSO concentrations, as reported by Banerjee et al. [130], that would affect the solvation of ferric ions.

The abovementioned studies provide evidence that many considerations must be accounted for in the use of cosolvents. Firstly, the gel matrix structure could become significantly influenced by different factors. Either by using physically crosslinked matrices, such as gelatin, or by using covalently linked macromolecules, such as polyvinyl alcohol with glutaraldehyde, it is the interaction between the solvent or mixture of solvents that determines the final structure, mesh size, and pore size in the gel. Specifically, hydrogen bonds between the solvent and the gel matrix, surface tension effects, and the hydrophobic/hydrophilic nature of the solvent mixture play an essential role in the mechanical properties of the gel. As a consequence of replacing a percentage of water with an organic solvent, or by changing the pH of the aqueous solution, the gel mechanical properties could change together with the solvation processes and diffusivity of the reactive species, thereby altering the reaction within the dosimetry system. For example, Sandrin et al. [131] studied the pore size of polyacrylamide crosslinked networks and the diffusion of an organic molecule, and observed that using a phosphate buffer solution with a pH of 10 caused an increase in the network pore size from 11 ± 1 nm to 38 ± 1 nm and reported a lower diffusion coefficient of an organic tracer in water than in the buffer. The authors also compared several theoretical models, proving that an attraction shell and repulsive interactions between the matrix and guest molecule were essential to describe the experimental values, indicating that the transport of organic species within a gel network is not only controlled by hindrance effects, but also by physical and chemical interactions among the components in polymer gel dosimetry.

### 7.3. Antioxidants, Oxygen Scavengers, and Inhibitors

Due to the nature of free radical chemistry, polymer gel dosimeters are susceptible to atmospheric and dissolved oxygen because it inhibits the polymerization processes. As a result, these gel dosimeters originally had to be manufactured in an oxygen-free environment by using nitrogen or argon, commonly known as anoxic dosimetry systems.

Fong et al. [34] developed a different type of polymer gel dosimeter, known as MAGIC, in which a metallo-organic complex was used to bind atmospheric oxygen, allowing a notable simplification in the manufacturing of PGDs. An aqueous solution of gelatin, open to the atmosphere, was mixed with methacrylic acid, copper(II) ions, ascorbic acid, and hydroquinone. Ascorbic acid bound the free oxygen contained within the aqueous gelatin matrix into metallo-organic complexes in a process initiated by copper sulfate [132]. Numerous authors subsequently reported on the manufacture of normoxic gels by using different antioxidants. At this point, it is important to highlight that antioxidants not only scavenge oxygen, but can also intervene in several reactions modifying the dose–response of dosimeters. Therefore, it is essential to study their effects on the properties of the resulting PGDs. For example, Khan et al. [133] demonstrated that using different concentrations of ascorbic acid as an oxygen scavenger in MAGIC exerted a strong impact on the dose rate dependence measured by the absolute sensitivity change over the whole dose rate interval. The results showed that the change in the absolute values of sensitivity with dose rate was significantly lower for higher oxygen scavenger concentrations. THPC was proposed as an effective oxygen scavenger and has been widely used in many dosimetry formulations. De Deene et al. [35] found that replacing ascorbic acid in MAGIC with THPC caused an increase in the R2 response of this gel. THPC could react with amine groups in the gelatin matrix to increase the gelatin crosslinking and coagulation [134], and, for example, reduce the gelation time [135,136]. In addition, Sedaghat et al. [137] demonstrated that THPC reduced the dose–response of acrylamide-based PGDs because it significantly affected the gelatin matrix, preventing the polymerization reaction.

Antioxidants can react with other radicals in the PGDs before or during irradiation, thereby modifying the radiation dose–response [138]. On one hand, too much oxygen could fully inhibit polymerization. On the other hand, unreacted antioxidant molecules, after reacting with oxygen, will have an impact on the polymerization reaction and radiation dose–response of the dosimeter because they can react with either water-free radicals, polymer radicals, or both. The amount of unreacted antioxidant molecules depends on the number of oxygen molecules penetrating the polymer gel dosimeter during manufacture and storage, an aspect that can compromise the accuracy and reproducibility of dosimetry measurements with normoxic polymer gels.

Polymerization inhibitors have been used and studied within polymer gel dosimeters to influence the efficiency of the polymerization and control the overall dose–response by altering the kinetics of the process. For this purpose, Magugliani et al. [139] evaluated the use of CuCl_3_, nitrobenzene, hydroquinone, and p-nitrophenol to quantitatively regulate the range of the response and sensitivity of PAGAT without detrimental effects on fundamental dosimetry quantities, such as the dose resolution, precision, and chemical stability. The authors reported a superior performance of *p*-nitrophenol in extending the dose range of PAGAT up to approximately 25 Gy, with no detrimental effects on the precision and dose resolution under MRI or optical analysis. The inhibition of phenols is a chain transfer to the inhibitor by means of OH addition and H atom abstraction with hydroxyl radicals and further reactions with OH•, HO_2_•, and O_2_, forming less reactive radicals or stable species, such as hydroquinone, benzoquinone, etc. [140], that also act as radical inhibitors.

### 7.4. Nanoparticles

In the search for effective approaches to meet the demands of human health, great efforts have been made in the development of nanomedicine able to screen the response of a specific treatment in real-time. In this scenario, researchers are working on the development of theranostic formulations, which are defined as multifunctional platforms that combine therapeutic and diagnostic properties. Some inorganic nanomaterials have intrinsic properties that can be used for their detection in the human body; therefore, their application in nanomedicine provides diagnostic capabilities. Quantum dots, graphene, carbon nanotubes, magnetic nanoparticles, and gold nanoparticles are the most frequent materials tested in the development of theranostics systems.

Nowadays, the integration of inorganic nanoparticles in PGDs is aimed at extending the concept of theranostics to external radiotherapy by using X-ray fluorescence and secondary electron emission of metallic nanoparticles as dose enhancers, which also serve to improve the dose delivery in the treatment of cancer. Furthermore, research should consider the characterization of the properties of these novel dosimeters considering the dose enhancement produced by X-ray fluorescence and secondary electrons, both Auger and Coster-Kronig [141,142]. Incident X-ray beams with photons capable of exciting the high-*Z*-absorption edges (K-, L-, M-, etc.) may produce photoelectric interactions, thus inducing internal transitions/relaxations leading to the emission of characteristic photons (fluorescent X-rays) or secondary electrons [143,144]. Following the initial ionization, a vacancy is created for the (free) quantum state that leaves the ionization (ejected) electron. Auger electron and fluorescent photon emissions are competitive decay or transition processes. Then, the short range of the emitted electrons suggests that local dose enhancements proceed. Some of the main processes involved in the interaction of radiation with high-*Z* materials and nanoparticles are depicted in Figure 6.

Different types of inorganic nanoparticles used in polymer dosimetry were reviewed by Titus et al. up to 2016 [146]. The research in this area focused on achieving the highest increment in the absorbed dose produced by the nanoparticles. In addition to the chemical nature of nanoparticles, there are other factors that influence the dose enhancement factor (*DEF*), such as the size and concentration of nanoparticles, and incident energy [147]. Gold nanoparticles (AuNPs) are one of the most studied materials as radiosensitizers to increase the efficiency of radiotherapy due to their high atomic number (*Z* = 79) and the effect of secondary electrons generated because of their interaction with radiation within the tissues. Recently, AuNPs with different sizes (10 nm and 100 nm) and concentrations (0.1, 0.2, 0.3, 0.4, and 0.5 mM) were tested in a methacrylic acid/gelatin dosimeter under the same conditions of brachytherapy treatment using an ^192^Ir radioisotope [148]. The investigation showed that the size of AuNPs influenced the *DEF*, and higher concentrations of nanoparticles yielded larger dose enhancement factors. Behrouzkia et al. [149] studied AuNPs with concentrations of 30 nm, 50 nm, and 100 nm in MAGICA, finding that the maximum *DEF* was achieved for 50 nm GNPs. It was assumed that the lower *DEF* of 100 nm AuNPs in comparison with 50 nm particles could be due to the self-absorption effect with larger diameters. Lima et al. [150] established that the higher the radiation dose, gold mass percentage, and/or intrinsic sensitivity of the dosimetry material, the higher the production of radiation-induced free-radicals, enhancing the probability of radical recombination and resulting in lower *DEF* values.

With regard to silver nanoparticles (AgNPs), similar to AuNPs, they can be used for dose enhancement in radiation therapy, but, unlike gold, they have shown antitumor activity. The synthesis of AgNPs is more versatile than AuNPs and they can even be synthesized by the reduction of silver nitrate with the amine groups of the gelatin matrix, which is already part of most of PGDs [151]. Vedelago et al. [152] reported the dose enhancement produced by the presence of AgNPs in Fricke gel dosimeters by measuring the difference in the absorbance of dosimeters irradiated above and below the silver K-edge. However, specific care must be taken with the addition of active species, such as metallic silver. For example, Merkis et al. [153] reported that THPC reacts with the precursor of silver nanoparticles (silver nitrate) to form AgCl precipitates and proposed using tetrakis(hydroxymethyl) phosphonium sulfate instead (THPS). The authors were able to assess the feasibility of using AgNPs in the nMAG dosimetry system.

Recently, Soliman et al. [154] proposed a different strategy involving AgNPs—they studied a hydrogel dosimeter based on polyacrylamide by using silver nitrate as a radiation-sensitive material and demonstrated that radiation induced the synthesis of AgNPs in the hydrogel. The formation of AgNPs was monitored by UV-Vis spectrophotometry within a dose range of 0 to 100 Gy. The results showed that the presence of glycerol in the matrix improved the radiation sensitivity, which could be explained considering that the free radical species generated from glycerol upon irradiation can serve as a reducing agent for silver ions (Ag^+^). In addition, the radiation sensitivity increases with increasing Ag^+^ concentrations up to 100 mM, and the response decreases at 150 mM. Nevertheless, it is worth considering that the absorption peak used to measure AgNP formation represents the local surface plasmon resonance (LSPR), which involves the resonant oscillation of conduction electrons. Therefore, the intensity of this band cannot be directly used as the result of a common absorbance phenomenon. In other words, the intensity of this peak can be related to the amount of AgNPs formed only if the AgNPs are spherical and homogeneously distributed, with a very narrow size distribution. According to the Mie theory, increasing the size of spherical AgNPs results in a red shift in the SPR peak position [155]. Therefore, to ensure that the correlation between the intensity of the LSRP and the AgNP amount is accurate, the size distributions of the AgNPs formed within dosimeters under different experimental conditions should be evaluated by, for example, electron microscopy. Merkis et al. [156] also observed that the intensity of the LSPR band in irradiated samples increased with the concentration of silver nitrate. However, regarding the size of the formed AgNPs, the authors reported a shift to a higher wavelength in the LSPR peak with an increase in the irradiation dose. Another common dose-enhancer and contrast agent is Gadolinium (*Z* = 64)—Gadolinium formulations have been commonly used as contrast agents for MRI imaging and they can also be used as a dose-enhancer [146,157]. Moreover, bismuth (*Z* = 83) can be used for theranostics, as it has a high electron density, low toxicity, and low cost compared with gold [158]. Furthermore, bismuth gadolinium oxide (BiGdO_3_) nanoparticles were tested as a radiosensitizer in radiation therapy and imaging by either X-ray CT or MRI by preparing a MAGIC gel dosimeter in the presence of BiGdO_3_ nanoparticles modified with polyethylene glycol for better biocompatibility. The authors verified the effect of the nanoparticles with gel dosimetry through in vitro and in vivo analyses, reporting sensitizer enhancement ratios of 1.75 and 1.66 with concentrations of 0.1 mg/mL in MCF-7 and 4T1 cell lines, respectively [159]. In this case, both bismuth and gadolinium provided CT contrast, while gadolinium could also be used for MRI contrast enhancement. Figure 7 presents some applications and relevant examples of dose and contrast enhancement by metallic nanoparticles.

## 8. Concluding Remarks

In this review, some of the most relevant interactions that could be present in novel gel dosimetry were discussed from a physicochemical perspective, aiming to contribute with available interpretations of frequent empirical analyses in the literature based on the fundamentals involved in Fricke gel dosimetry and polymer gel dosimetry, and particularly on the underlying chemical processes in the dosimetry system.

Additionally, some novel formulations and alternative gel dosimetry systems from the last five years were addressed throughout the discussion as a milestone in the application of innovative gel dosimetry in clinical applications.

## Figures and Tables

**Figure 1 gels-08-00663-f001:**
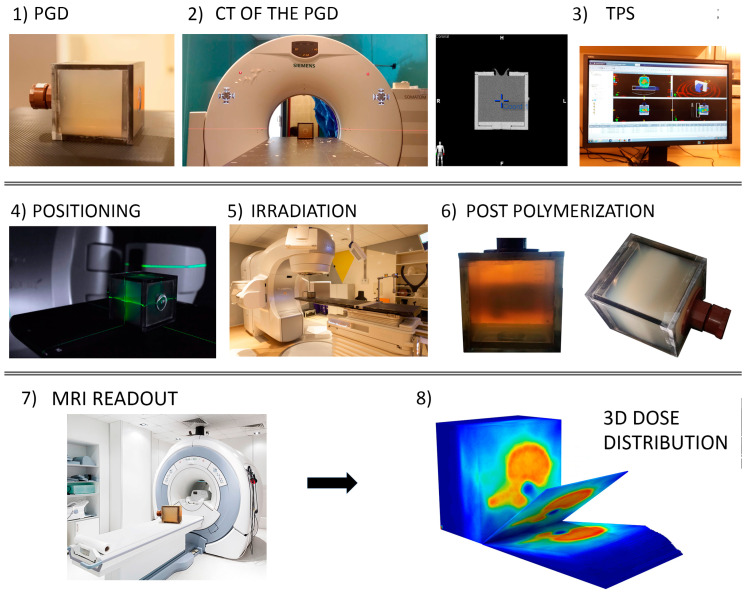
Typical end-to-end application of gel dosimetry. (**1**) Gel dosimetry in a cubic phantom (extension to anthropomorphic phantoms is straightforward), (**2**) pre-irradiation CT scanning (**3**) treatment planning, (**4**) positioning, (**5**) irradiation, (**6**) 3D dosimetry response (post-polymerization processes present only in PGD), (**7**) readout of the dosimetry, (**8**) final 3D dose distribution generated from the calibration and used gel dosimetry. (Reproduced and adapted with permission from reference [3]). PGD: Polymer Gel dosimetry; TPS: Treatment Planning System; MRI: Magnetic Resonance Imaging; CT: Computed Tomography.

**Figure 2 gels-08-00663-f002:**
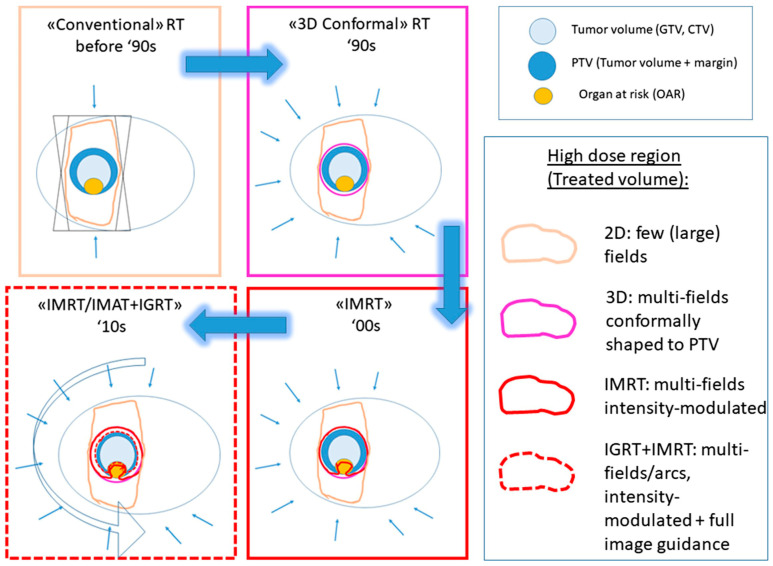
Scheme of a PTV next to healthy tissue treated with different radiotherapy treatments to represent the evolution of dose delivery in radiotherapy over the last 25 years. (Reproduced with permission from reference [4]). CTV: clinical target volume; GTV: gross target volume; IGRT: image guided radiotherapy; IMAT: intensity modulated arch therapy; IMRT: intensity modulated radiotherapy; PTV: planning target volume; RT: radiotherapy.

**Figure 3 gels-08-00663-f003:**
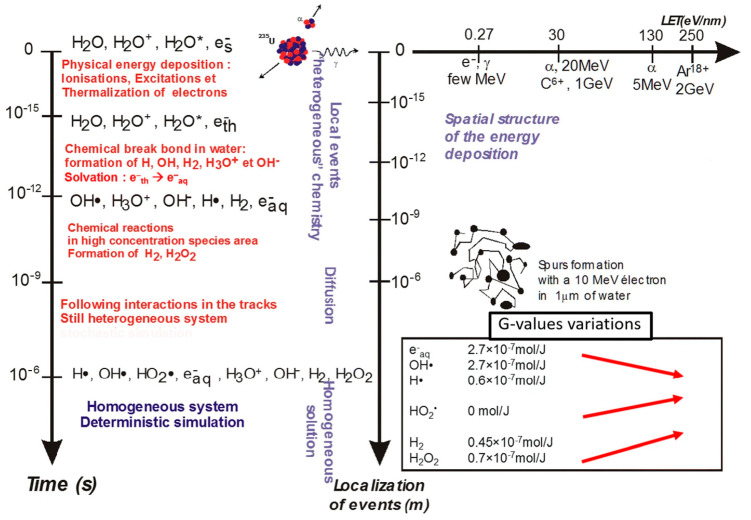
Representation of the reactions of transient species produced by irradiation in pure water. Time is represented on the vertical left axis, localizations in space on the central vertical axis, and *LET* on the horizontal axis of the right scheme. The values of primary radiolytic yields (*G*-values) are tabulated and shown with respect to the *LET* axis (from the lowest *LET* to the highest one). Red arrows indicate the typical variations in the *G*-values when *LET* increases. (Adapted with permission from reference [78]). H_2_O* represents excited water.

**Figure 4 gels-08-00663-f004:**
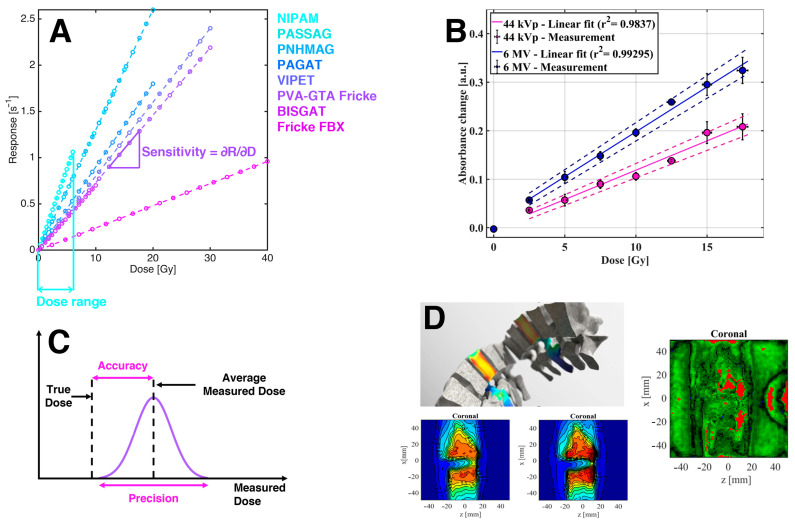
Relevant dosimetry parameters. (**A**) Qualitative sensitivity and dose range values for different gel dosimetry systems. (**B**) Comparison of the response behavior at two very different energy values for Fricke gel. (**C**) Graphical representation of the accuracy and precision of dosimetry measurements. (**D**) Typical quality assurance analysis with gel dosimetry. Example of an irradiation of a vertebra (**upper left**), dose map for a coronal slice obtained with TPS (**lower left**), dose map for a coronal slice obtained with a PGD (**lower center**), and 2D gamma pass map between the TPS and PGD dose maps (**right**).

**Figure 5 gels-08-00663-f005:**
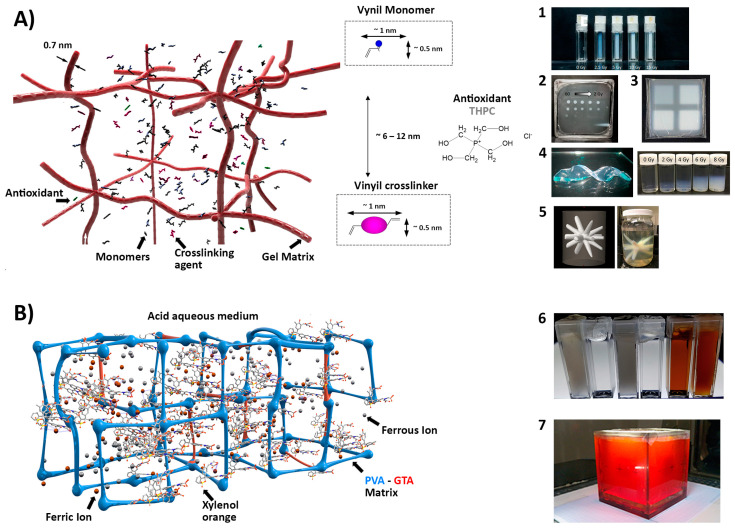
Schematic representation of gel dosimetry based on a gel matrix. (**A**) Polymer gel dosimetry scheme along with the typical dimensions for vinyl monomers. (**1**) PAGAT in a gelatin matrix, prepared in spectrophotometric cuvettes irradiated at different doses up to 15 Gy; (**2**) PAGAT in a gelatin matrix prepared in a layer-type container irradiated with circular collimation and different doses up to 60 Gy; (**3**) MAGT-A in a gelatin and agar matrix prepared in flexible film containers irradiated with squared collimation with doses up to 8 Gy (Reproduced with permission from reference [122]); (**4**) Aquajoint^TM^ matrix (**left**) and VIPET^AQUA^ irradiated up to 8 Gy (**right**) (Reproduced with permission from reference [123]); (**5**) NIPAM in a gelatin matrix irradiated in an isocenter verification plan (**left**) and the TPS 3D rendering (**right**) (Reproduced with permission from reference [43]). (**B**) PVA-GTA Fricke gel dosimetry scheme. (**6**) Different PVA-GTA FGDs; from left to right: plain cryo-gel, plain PVA-GTA gel, Fricke cryo-gel, Fricke PVA-GTA gel, Fricke PVA-GTA gel with xylenol orange, and Fricke cryo-gel with xylenol orange (Reproduced with permission from reference [119]); (**7**) Fricke gel dosimetry in a gelatin matrix in a 1 L cubic container.

**Figure 6 gels-08-00663-f006:**
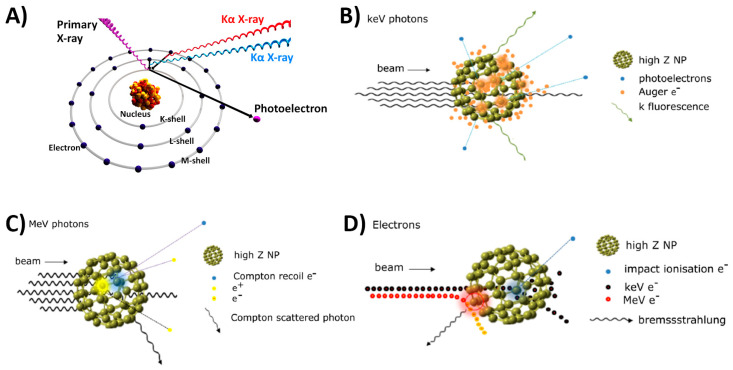
Principle and application of gel metallic nanoparticles for theranostics. (**A**) Atomic interaction of X-rays and generation of K-shell characteristics fluorescence. Schematic illustration of inelastic interactions with a high-*Z* nanoparticle for (**B**) incident keV photons (orange clouds represent photoelectric events), (**C**) incident MeV photons (blue and yellow clouds represent Compton scatter and pair production events, respectively), and (**D**) incident electrons (blue and red clouds represent large- and small-impact parameters leading to ionization and Bremsstrahlung, respectively). (**B**–**D** figures are reproduced with permission from reference [145]).

**Figure 7 gels-08-00663-f007:**
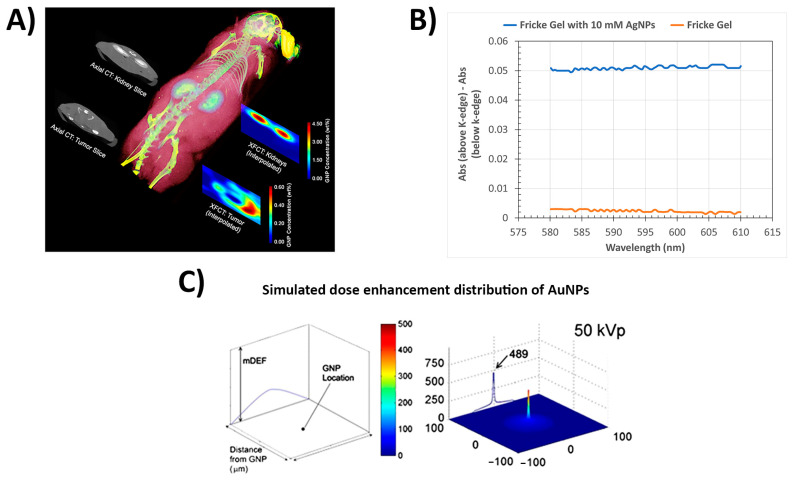
Application of high-*Z* nanoparticles as contrast agents and dose enhancement measurements. (**A**) Example of gold nanoparticles for contrast enhancement illustrating the possibility of quantitative multimodal imaging of AuNP distributions in an animal using X-ray fluorescence computed tomography (XFCT) and conventional CT together (reproduced with permission from reference [160]. (**B**) Experimental dose enhancement in Fricke gel dosimetry with AgNPs irradiated with energies below and above the k-edge of silver. (**C**) Spatial variation in mDEF around a AuNP. mDEF represents the factor by which the dose would be increased by replacing a hypothetical water nanoparticle with a AuNP. The results are shown along the radial direction from a AuNP at the center. (Adapted and reproduced with permission from reference [161]).

**Table 1 gels-08-00663-t001:** Included clinical studies with polymer gel dosimetry published since 2017.

Author and Year	Gel Type	Beam Type	Treatment Type	Technique Type	Readout System
Yao et al. 2019 [36]	NIPAM	Photon	External	IMRT and VMAT	Optical CT
Elter et al., 2019 [37]	PAGAT	Photon	External	MRgRT	MRI
Hillbrand et al., 2019 [38]	VIP	Proton	External	Pencil beam	MRI
Watanabe et al., 2019 [39]	VIPET	^192^Ir	Brachytherapy	-	MRI
Abtahi et al., 2020 [40]	NIPAM	Photon	External	Intraoperative radiotherapy	MRI
Chou et al., 2020 [41]	NIPAM	Photon	External	IGRT	Optical CT
Kozicki et al., 2020 [42]	VIC and VIC-T	Photon	External	Stereotactic radiosurgery	MRI
Pant et al., 2020 [43]	NIPAM	Photon	External	Stereotactic radiosurgery	CBCT
Mann et al., 2020 [44]	PAGAT	Photon	External	IGRT	MRI
Schwahofer et al., 2020 [45]	PAGAT	Photon	External	MRgRT	MRI
Elter et al., 2021 [46]	PAGAT	Photon	External	MRgRT	MRI
Nezhad et al., 2021 [47]	MAGAT	Photon	External	Intraoperative radiotherapy	MRI
Azadeh et al., 2022 [48]	MAGIC-f	Photons	External	Stereotactic radiosurgery	MRI
Fuse et al., 2022 [49]	PAGAT-MgCl_2_	Photon	External	IMRT	MRI
Kim et al., 2022 [50]	MAGAT	Photon	External	MR-Linac (isocenter verification)	MRI
Kudrevicius et al., 2022 [51]	nPAG	Photon	External	Gamma knife	MRI
Watanabe et al., 2022 [52]	VIPET	^192^ Ir and photon	Brachytherapy and external	- IMRT	MRI

## Data Availability

Not applicable.

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
