# Peer review of "Chemical Overview of Gel Dosimetry Systems: A Comprehensive Review"

_gels, 2022, doi:10.3390/gels8100663_

Round 1

Reviewer 1 Report

An intriguing review by Macchione et al. on gel dosimetry's fundamentals and chemical descriptions. Particularly in the research of materials and formulations since 2017. The report is exhaustive and comprehensively written. Therefore, authors should consider certain points.

1)     L22: pls amend: “this revision...”

2)     Title suggestion: Chemical overview of gel dosimetry systems: a comprehensive review

3)     The number of keywords is excessive; please choose the five most relevant.

4)     Since you mentioned 2017 in your review. It would be helpful if you could provide a method (or diagram) describing how you extracted references for your comprehensive review.

5)     Using our own graph, could you describe the progress or history of gel dosimetry research over time? This would allow the reader to comprehend the research trends in gel dosimetry.

6)     Please define the acronym first. Please check all; for example, IGRT and MRgRT

7)     There are several grammatical errors. The authors should double-check

8)     The majority of sentences in the paper lack citations. If you have cited sentences from other sources, please include proper citations (Please see in the introduction and Brief history of gel dosimetry)

Reviewer 2 Report

This mauscript is well organized and the subject of gel dosimetry is comprehensively compiled.

Author Response

Authors are grateful for the valuable and positive comments.

Reviewer 3 Report

Dear Editor, dear Authors, Micaela Macchione et al., submitted a review paper on gel dosimetry systems widely used in theranostics. The authors main aim from this manuscript is to provide some fundamentals and chemical descriptions relevant in gel dosimetry, and particularly in materials and formulations proposed during the last five years. Therefore, they have first introduced the concept of radiation therapy that paved the road toward the main idea of the review on gel dosimetry. After a historical part and read out techniques, the authors mainly discussed the most relevant interactions that could be present in novel gel dosimetry and their main properties and applications by selecting examples published in the last five years. For my opinion, the manuscript is of interest, and well written. The references and figures are well selected and discussed. Therefore, I believe that the manuscript can be accepted for publication in its present format for publication in Gels MDPI Journal. I have very minor comments to the authors that should be revised.

Comments to the authors

-          Page 1 lines 35-40. I suggest that the authors include some references here especially that they are listing a series of different mechanisms of interaction of electromagnetic radiation with matter.

-          Page 1, line 42 “the corresponding double 41 differential cross section (d?/dEd?) that depends on the kinetic energy (E) of the particle 42 passing through.” The authors should define ? and ? here to clarify. Please revise.

Sincerely yours,

Reviewer 4 Report

I do not feel that the introduction fits well in the review. I would expect the introduction to describe an overview of the use of gel dosimeters, both in research and medical applications. This is missing in the paper. A short and concise overview of the use of gel dosimetry is missing. For example, lines 34 to 51 is a description of the interaction of radiation with matter. It is not exactly an introduction to gel dosimetry. Also, from 89 to 102 where we can find an excellent description on the radical production… All of this is important and should be described as it is, but, in my opinion, it must have a designated section.

I understand that the majority if the images were adapted but in most cases they should be improved in quality.

In geral, the references are adequate and recent and the paper is well written and well structured.

Round 2

Reviewer 1 Report

My suggestions were integrated into the final manuscript by the authors.

I am also satisfied with the revised manuscript. Thus, the final manuscript is acceptable to be published in Gels.